# Solving the Fuzzy Job Shop Scheduling Problem via Learning Approaches

## Abstract

The fuzzy job shop scheduling problem (FJSSP) emerges as an innovative extension to the conventional job shop scheduling problem (JSSP), incorporating a layer of uncertainty that aligns the model more closely with the complexities of real-world manufacturing environments. This enhancement, while enhancing its applicability, concurrently escalates the computational complexity of deriving solutions. In the domain of traditional scheduling, neural combinatorial optimization (NCO) has recently demonstrated remarkable efficacy. However, its application to the realm of fuzzy scheduling has been relatively unexplored. This paper aims to bridge this gap by investigating the feasibility of employing neural networks to assimilate and process fuzzy information for the resolution of FJSSP, thereby leveraging the advancements in NCO to enhance fuzzy scheduling methodologies. To this end, we present a self-supervised algorithm for the FJSSP (SS-FJSSP). This algorithm employs an iterative mechanism to refine pseudo-labels, progressively transitioning from suboptimal to optimal solutions. This innovative approach adeptly circumvents the significant challenge of procuring true labels, a common challenge in NCO frameworks. Experiments demonstrate that our SS-FJSSP algorithm yields results on a par with the state-of-the-art methods while achieving a remarkable reduction in computational time, specifically being two orders of magnitude faster.

## 1 Introduction

The job shop scheduling problem (JSSP) is a well-established combinatorial optimization problem (COP) that holds both theoretical significance and practical relevance (Zhang et al., 2024). The traditional JSSP describes the processing time in the form of crisp number. However, in real-world manufacturing scenarios, numerous uncertain factors, such as human variability (He et al., 2021) and machine flexibility (Huang et al., 2024), often preclude the accurate specification of processing times. To overcome this limitation, the fuzzy JSSP (FJSSP) has emerged and is attracting increasing attention in the field (Abdullah & Abdolrazzagh-Nezhad, 2014; Lin, 2002). Specifically, in the FJSSP, processing times are represented as fuzzy numbers. These uncertainties diminish the practical applicability of the JSSP.

The existing algorithms for FJSSP are mainly heuristic algorithms (Gendreau & Potvin, 2005). Li et al. (2023) developed a bi-population balancing multiobjective evolutionary algorithm for distributed flexible FJSSP. Gao et al. (2020) devised a differential evolution algorithm with an innovative selection mechanism to to more effectively tackle FJSSP. Li et al. (2020) engineered an enhanced artificial immune system algorithm to address flexible FJSSP. Sun et al. (2019) crafted an effective hybrid cooperative coevolution algorithm aimed at minimizing the fuzzy makespan of flexible FJSSP. The integration of particle swarm optimization and genetic algorithms significantly bolstered the convergence capabilities of the proposed algorithm. Wang et al. (2022) utilized fuzzy relative entropy to transform a multiobjective optimization FJSSP into a single-objective optimization problem and designed a hybrid adaptive differential evolution algorithm to resolve it. Pan et al. (2021) concentrated on the energy-efficient flexible FJSSP and developed a bi-population evolutionary algorithm with feedback mechanisms to address it effectively.

The surge in deep learning has catalyzed the emergence of neural combinatorial optimization (NCO), sparking a burgeoning interest in leveraging learning-based approaches to solve combinatorial op-

timization problems (COPs) (Bengio et al., 2021; Chen & Tian, 2019; Falkner et al., 2022). Initial forays into this domain focused on foundational COPs. Vinyals et al. (2015) were trailblazers with the pointer network, a novel neural network architecture specifically designed for the traveling salesman problem (TSP), thereby underscoring the potential of neural networks in addressing COPs. Bello et al. (2016) combined neural networks with reinforcement learning for the TSP, achieving near-optimal results. This method, while computationally demanding, reduced reliance on intricate engineering and heuristic design. Kool et al. (2018) further refined the pointer network by incorporating attention mechanisms, resulting in significant enhancements in the performance of both the TSP and the vehicle routing problem (VRP). Nazari et al. (2018) introduced a comprehensive framework that utilized reinforcement learning for the VRP, surpassing traditional heuristics on medium-scale capacitated VRP instances without a substantial increase in computational time. The application of NCO has since expanded into the realm of scheduling problems. Zhang et al. (2020) developed a deep reinforcement learning model that autonomously learned JSSP priority dispatch rules. They also introduced a graph neural network (GNN) for encoding states, outperforming existing dispatch rules. Kotary et al. (2022) presented a deep learning strategy that yielded efficient and accurate JSSP approximations; they integrated Lagrangian duality to manage problem constraints, showcasing the potential for generating high-quality JSSP solutions with minimal computational overhead. Corsini et al. (2024) explored a self-supervised training strategy for JSSP, training generative models on multiple solution samples and using the optimal solution as a pseudo-label, thus eliminating the need for costly ground-truth solutions and overcoming challenges associated with supervised learning. However, all these studies are predicated on deterministic environments, and to date, no research has addressed the fuzzy scheduling problem using learning approaches. In this paper, we aim to investigate whether neural networks can assimilate fuzzy information and apply it to solve the FJSSP, thereby integrating the advancements of NCO into fuzzy scheduling.

The main contributions of this work are as follows:

- We propose a self-supervised algorithm for FJSSP (SS-FJSSP), which can learn fuzzy information and solve fuzzy scheduling problems in a learning approach.
- We employ an iterative refinement process to incrementally transform initially inaccurate labels into authentic ones, effectively addressing the challenge of acquiring true labels in the realm of NCOs.
- Experimental results indicate that the SS-FJSSP algorithm matches the performance of the state-of-the-art methods and significantly reduces computation time, with speeds up to 100 times faster.

## 2 PRELIMINARIES

### 2.1 FUZZY NUMBER

Fuzzy numbers are used to represent the processing time for fuzzy scheduling, which is described in this section.

In manufacturing environments, precise processing times are often elusive due to variables such as the diverse skill levels of workers (Shao et al., 2024). While exact durations may not be predictable, experts can often draw on historical data to provide estimated durations (Itoh & Ishii, 1999). To address this unpredictability, a prevalent strategy involves estimating within confidence intervals. When certain values are more likely, opting for a fuzzy interval or number becomes an appropriate choice (Fortemps, 1997).

Assume $S$ is a fuzzy set defined on $\mathbb{R}$, with a membership function $\mu_S : \mathbb{R} \to [0, 1]$. The $\alpha$-cut of $S$ is defined as $S_\alpha = \{x \in \mathbb{R} : \mu_S(x) \geq \alpha\}$, $\alpha \in (0, 1]$, and the support is $S_0 = \{x \in \mathbb{R} : \mu_S(x) > 0\}$. A fuzzy interval is delineated by its $\alpha$-cuts being confined, and a fuzzy number $\tilde{N}$, with a compact support and a pronounced modal value, is depicted by closed intervals $\tilde{N}_\alpha = [\underline{n}_\alpha, \bar{n}_\alpha]$.

The triangular fuzzy number (TFN) (Zhu et al., 2020) is frequently utilized in fuzzy scheduling problems. Let $\tilde{A}$ be a TFN denoted as $\tilde{A} = (a_1, a_2, a_3)$, where $a_1$ and $a_3$ outline the range of potential values and $a_2$ signifies the modal value within this range. The membership function of $\tilde{A}$ is articulated as follows:

$$\mu_{\tilde{A}}(x) = \begin{cases} \frac{x-a_1}{a_2-a_1}, & \text{if } a_1 < x \le a_2, \\ \frac{a_3-x}{a_3-a_2}, & \text{if } a_2 < x < a_3, \\ 0, & \text{otherwise.} \end{cases} \tag{1}$$

Let $\tilde{A} = (a_1, a_2, a_3)$ and $\tilde{B} = (b_1, b_2, b_3)$ represent two TFNs. The operations on these TFNs are defined as follows:

**1. Additional Operation.** According to (Nguyen et al., 2018), the sum of $\tilde{A}$ and $\tilde{B}$ is as follows:

$$\widetilde{A} + \widetilde{B} = (a_1 + b_1, a_2 + b_2, a_3 + b_3). \tag{2}$$

**2. Max Operation.** According to (Lei, 2010a), the max operation of $\tilde{A}$ and $\tilde{B}$ is as follows:

$$\max(\tilde{A}, \tilde{B}) = \begin{cases} \tilde{A}, & \text{if } \tilde{A} \ge \tilde{B}, \\ \tilde{B}, & \text{otherwise.} \end{cases} \tag{3}$$

This operation is grounded in a ranking method, and for this paper, we employ the method proposed by (Heilpern, 1992). This method defuzzifies TFNs to crisp numbers, allowing the comparison of these values to determine the relationship between the TFNs, as shown in the following formula:

$$\text{Defuzz}(\tilde{A}) = \frac{a_1 + 2a_2 + a_3}{4}. \tag{4}$$

In addition to the above well-defined operations, this paper also deals with subtraction, division, and other operations on TFNs. Therefore, the defuzzification method defined in Eq. (4) is also applied to these operations, ensuring that the neural network can effectively learn and process the relevant information.

## 2.2 FJSSP

The FJSSP (Vela et al., 2020) involves a collection of jobs, machines, and operations. Specifically, there are $n$ jobs denoted by set $\mathcal{J}$, $m$ machines represented by set $\mathcal{M}$, and $N$ operations within set $\mathcal{O}$. Each operation $i \in \mathcal{O}$ is associated with a unique job $J_i \in \mathcal{J}$, processed by a specific machine $M_i \in \mathcal{M}$, and has an uncertain processing time denoted by a TFN $\tilde{t}_i$. The operations are linked by a binary relationship $\rightarrow$ that forms chains for each job. If operation $i$ precedes $j$ ($i \rightarrow j$), they share the same job $J_i = J_j$, and no other operation $x$ can exist such that $i \rightarrow x$ or $x \rightarrow j$. Let $\mathcal{S}$ be the set of scheduling scheme; The objective of FJSSP is to minimize the fuzzy makespan, i.e., to find the fuzzy start time $\tilde{s}_i$ for each operation $i \in \mathcal{O}$ to minimize the following objective over all possible schemes:

$$\max_{i \in \mathcal{O}} \tilde{s}_i + \tilde{t}_i \tag{5}$$

$$\text{s.t.} \quad \tilde{s}_i \ge 0, \quad \forall i \in \mathcal{O}, \tag{6}$$

$$\tilde{s}_j \ge \tilde{s}_i + \tilde{t}_i, \quad \text{if } i \rightarrow j, i, j \in \mathcal{O}, \tag{7}$$

$$\tilde{s}_j \ge \tilde{s}_i + \tilde{t}_i \wedge \tilde{s}_i \ge \tilde{s}_j + \tilde{t}_j, \quad \text{if } M_i = M_j, i, j \in \mathcal{O}. \tag{8}$$

To enable the neural network to assimilate the intricate constraint information inherent in the FJSSP, this paper deviates from the conventional mixed-integer linear programming models typically utilized in heuristic algorithms, as referenced by (Tirkolaee et al., 2020). Instead, we adopt the disjunctive graph approach, as introduced by (Van Laarhoven et al., 1992), to effectively model the FJSSP. This methodological choice facilitates a more nuanced representation of the scheduling constraints, enhancing the capacity of network to learn and optimize solutions within this complex domain.

The disjunctive graph $G = (V, A, E)$ characterizes problems as follows:

- $V = \mathcal{O} \cup \{S, T\}$, where $S$ and $T$ denote the starting and ending virtual nodes, respectively, each with a processing time of zero.
- $A$ encompasses ordered pairs $(i, j)$ for $i, j \in \mathcal{O}$ such that $i \rightarrow j$, along with pairs $(S, j)$ for the first operations of all jobs and $(i, T)$ for the last operations, representing the directed connections between operations within the jobs.

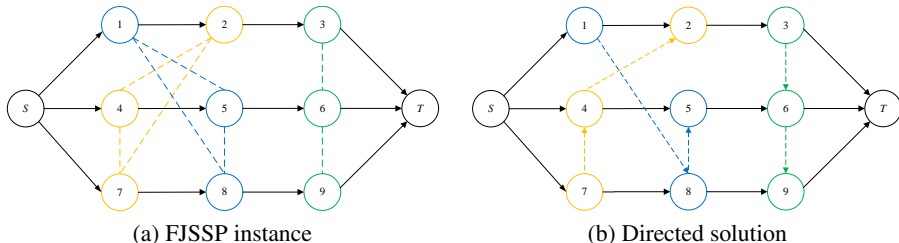

(a) FJSSP instance           (b) Directed solution

Figure 1: Disjunctive graph model. (a) illustrates a 3 × 3 FJSSP instance. The black solid lines delineate set $A$, while the colored dashed lines enclose set $E$. Operations with the same frame color must be executed on the same machine. (b) presents a solution for (a), where all undirected edges have been directed.

- $E$ includes pairs $(i, j)$ where $M_i = M_j$, indicating undirected connections between operations assigned to the same machine.

For each pair of operations $i, j \in \mathcal{O}$ with $i \to j$, the constraint in Eq. (6) is denoted as a directed edge $(i, j)$ in $A$. Similarly, for each pair of operations $i, j \in \mathcal{O}$ with $M_i = M_j$, the constraint in Eq. (7) is denoted as an undirected edge $(i, j)$ in $E$, and the two ways of solving the disjunction correspond to the two possible orientations of the edge. Therefore, finding a solution of FJSSP is equivalent to determining the direction of each undirected edge, resulting in a directed acyclic graph. An example of a disjunctive graph for an FJSSP instance and its solution are shown in Figure 1.

Moreover, since the directed edges give constraints on the processing order of all the operations in each job, in decision making, we only need to determine which job (rather than which operation) needs to be processed. For example, Figure 1(b) corresponds to a scheduling scheme $[1, 3, 2, 3, 1, 2, 1, 2, 3]$, and the objective of the algorithm proposed in this paper is to decide on scheduling schemes such as this one and make their fuzzy makespan as small as possible.

## 3 SS-FJSSP

The SS-FJSSP framework is comprised of two main components, namely an encoder and a decoder. The role of encoder is to assimilate and comprehend the FJSSP information from a holistic standpoint, capturing the broader context and constraints. Subsequently, the decoder leverages the insights gleaned by the encoder to make informed, sequential decisions, breaking down the complex scheduling problem into manageable steps. Moreover, it is essential to preprocess the features of the raw data prior to encoding to enhance the learning process. Each component is described in detail below.

### 3.1 Feature extraction and encoder

In the disjunctive graph model, the data information, specifically the fuzzy processing times, is stored at the vertices representing the operations, while constraints are defined by the edges. However, the raw data at these vertices, which includes only the individual fuzzy processing times, is insufficient for making accurate decisions. To achieve this, we require more comprehensive information. This is because, in addition to its own numerical magnitude, its role (relative numerical magnitude) in the job and machine to which it belongs is also important. To address this, we employ a feature vector $\boldsymbol{x_i} \in \mathbb{R}^{18}$, associated with operation $i$, to encapsulate the data information for the entire disjunctive graph. The feature vector $\boldsymbol{x}_i$ contains the following entries:

$$\tilde{t}_i = (t_1, t_2, t_3) \in \mathbb{R}^3, \tag{9}$$

$$\text{Defuzz}\left(\tilde{t}_i\right) \in \mathbb{R}, \tag{10}$$

$$\frac{\sum_{j=\text{St}(i)}^{i} \text{Defuzz}\left(\tilde{t}_j\right)}{\sum_{j=\text{St}(i)}^{\text{End}(i)} \text{Defuzz}\left(\tilde{t}_j\right)} \in \mathbb{R}, \tag{11}$$

$$\frac{\sum_{j=i+1}^{\text{End}(i)} \text{Defuzz}\left(\tilde{t}_j\right)}{\sum_{j=\text{St}(i)}^{\text{End}(i)} \text{Defuzz}\left(\tilde{t}_j\right)} \in \mathbb{R}, \tag{12}$$

$$\text{Quartile}\left(J_i\right) \in \mathbb{R}^3, \tag{13}$$

$$\text{Quartile}\left(M_i\right) \in \mathbb{R}^3, \tag{14}$$

$$\text{Defuzz}\left(\tilde{t}_i\right) - \text{Defuzz}\left(\text{Quartile}\left(J_i\right)\right) \in \mathbb{R}, \tag{15}$$

$$\text{Defuzz}\left(\tilde{t}_i\right) - \text{Defuzz}\left(\text{Quartile}\left(M_i\right)\right) \in \mathbb{R}, \tag{16}$$

where $\text{St}(i)$ and $\text{End}(i)$ denote the the first and last operation of the job that operation $i$ belongs, respectively. Quartile$(\cdot)$ calculates the quartiles. Eqs. (9) and (10) describe local information and represent the fuzzy processing time and the defuzzified processing time of $\mathcal{O}_i$, respectively. Other equations describe global information. Eqs. (11) and (12) describe how much of the job to which it belongs has been completed and how much is left after processing $\mathcal{O}_i$, respectively. Eqs. (13) and (14) describe the quartiles of fuzzy processing time for the job and machine to which $\mathcal{O}_i$ belongs, respectively. Eqs. (15) and (16) describe the difference of the defuzzified fuzzy processing time of $\mathcal{O}_i$ and Eqs. (13) and (14), respectively.

Subsequently, a two-layer Graph Attention Network (GAT) (Brody et al., 2021) is employed to extract and learn valuable information from the disjunctive graph, which transforms 18-dimensional $\boldsymbol{x_i}$ into a $h$-dimensional $\boldsymbol{e_i}$. The primary advantage of $\boldsymbol{e_i}$ over $\boldsymbol{x_i}$ is that it complements the relationship between edges. Furthermore, in order not to weaken the information of the vertices, at each layer of the GAT, the output is concatenated with the original feature vector $\boldsymbol{x_i}$. The formulation of encoder is detailed as follows:

$$\boldsymbol{e_i} = \left[\boldsymbol{x_i}\| \text{ReLU}\left(\text{GAT}_2\left(\left[\boldsymbol{x_i}\| \text{ReLU}\left(\text{GAT}_1\left(\boldsymbol{x_i}, G\right)\right)\right], G\right)\right)\right], \tag{17}$$

where "$\|$" is the concatenation operation.

## 3.2 Decoder

The decoder is composed of two parts: a state network and a decision network. The former is responsible for updating state, and the latter is responsible for making decisions based on the information provided by the decoder and memory network. The two networks are described below.

**1. The state network.** The SS-FJSSP makes decisions step by step, and each decision impacts the future. Therefore, after each decision, it is necessary to update the state and convey it to the SS-FJSSP to enhance the accuracy of subsequent decisions.

First, we generate an 11-dimensional context vector $\boldsymbol{c_i}$ ($i = 1, 2, \ldots, n$) for each job to describe its information. Assuming that $o_{t,i}$ denotes the ready operation of job $\mathcal{J}_i$ at step $t$, and its predecessor is $o_{t,i} - 1$. The context vector $\boldsymbol{c_i} \in \mathbb{R}^{11}$ contains the following entries:

$$\text{Defuzz}\left(\text{CT}(o_{t,i} - 1)\right) - \text{Defuzz}\left(\text{CT}(M_{o_{t,i}})\right) \in \mathbb{R}, \tag{18}$$

$$\frac{\text{Defuzz}\left(\text{CT}(o_{t,i} - 1)\right)}{\text{Defuzz}\left(\max_{i=1,\ldots,n} \text{CT}(\mathcal{J}_i)\right)} \in \mathbb{R}, \tag{19}$$

$$\text{Defuzz}\left(\text{CT}(o_{t,i} - 1)\right) - \frac{\text{Defuzz}\left(\sum_{i=1}^{n} \text{CT}(\mathcal{J}_i)\right)}{n} \in \mathbb{R}, \tag{20}$$

$$\text{Defuzz}\left(\text{CT}(o_{t,i} - 1)\right) - \text{Defuzz}\left(\text{Quartile}(\mathcal{J})\right) \in \mathbb{R}^3, \tag{21}$$

$$\frac{\text{Defuzz}\left(\text{CT}(M_{o_{t,i}})\right)}{\text{Defuzz}\left(\max_{i=1,\ldots,m} \text{CT}(\mathcal{M}_i)\right)} \in \mathbb{R}, \tag{22}$$

$$\text{Defuzz}\left(\text{CT}(M_{o_{t,i}})\right) - \frac{\text{Defuzz}\left(\sum_{i=1}^{m} \text{CT}(\mathcal{M}_i)\right)}{m} \in \mathbb{R}, \tag{23}$$

$$\text{Defuzz}\left(\text{CT}(o_{t,i} - 1)\right) - \text{Defuzz}\left(\text{Quartile}(\mathcal{M})\right) \in \mathbb{R}^3, \tag{24}$$

where $\text{CT}(\cdot)$ calculate the fuzzy completion time. Eq. (18) describes the relationship between two factors that affect the processing of $o_{t,i}$: whether the predecessor operation is complete and whether the required machine is idle. Eq. (19) measures how close the fuzzy completion time of $J_{o_{t,i}}$ is to

the current fuzzy makespan. Eq. (20) measures how early or late the fuzzy completion time of $J_{o_{t,i}}$ is compared to the average fuzzy completion time of all jobs in the current state. Eq. (21) describes the relative fuzzy completion time of $J_{o_{t,i}}$ w.r.t. other jobs. Eq. (22) measures how close the fuzzy completion time of $M_{o_{t,i}}$ is to the current fuzzy makespan. Eq. (23) measures how early or late the fuzzy completion time of $M_{o_{t,i}}$ is compared to the average completion time of all machines in the current state. Eq. (24) describes the relative fuzzy completion time of $M_{o_{t,i}}$ w.r.t. other machines.

Second, the context vectors are advancedly integrated through Eq. (25) to derive the state vectors $s_i \in \mathbb{R}^d (i = 1, 2, \ldots, n)$. These vectors serve as a pivotal input to the decision network, enabling it to make informed decisions effectively.

$$s_i = \text{ReLU}\left(\left[c_i \mathbf{W_1} + \underset{j=1,\ldots,n}{\text{MHA}}\left(c_j \mathbf{W_1}\right)\right]\mathbf{W_2}\right),\tag{25}$$

where $\mathbf{W_1}$ and $\mathbf{W_2}$ are learnable parameter matrices, and MHA denotes the multi-head attention layer (Vaswani, 2017).

**2. The decision network.** This network combines the $e_{o_{t,i}}$ generated by the encoder that contains global information about the FJSSP, and the $s_i$ generated by the memory network that contains local state information, to generate the probability of choosing a job for the current decision step. More specifically,

$$z_i = \text{FNN}\left(\left[e_{o_{t,i}} \| s_i\right]\right) \in \mathbb{R},\tag{26}$$

where FNN denotes the feedforward neural network (Glorot & Bengio, 2010). Then, the probabilities $p_i$ for processing the job $\mathcal{J}_i$ in the current step can be obtained by applying the Softmax function to $z_i$.

Next, a complete scheduling scheme can be generated based on the probability vector $p = [p_1, p_2, \ldots, p_n]^T$, denoting the probability that each job is selected. Specifically, first, at step 1, a job is randomly selected based on $p$ as the decision for this step. Then, the context vectors and state vectors are updated. Finally, based on the new context vectors and state vectors, the probability vector $p$ for next step is generated. Thus, an autoregressive process is formed until all jobs have completed. Moreover, if a job has completed, its probability is set to 0 by a mask operation at subsequent steps.

### 3.3 TRAINING STRATEGY

Since the FJSSP is an NP-Hard problem (Vela et al., 2020), obtaining true labels is impractical, and this results in the difficulty of solving this problem with a learning approach. Inspired by the work of (Corsini et al., 2024), we overcome this difficulty in the following ways. For each FJSSP instance, we generate $\alpha$ solutions through $\alpha$ parallel decision-making processes and select the appropriate solution as the pseudo-label to minimize the following loss function:

$$\mathcal{L}(\pi, \hat{\pi}) = -\frac{1}{mn}\sum_{t=1}^{mn}\log(p_{t,y_t}),\tag{27}$$

where $\pi$ and $\hat{\pi}$ denote the scheduling scheme of the pseudo-label and the predicted scheduling scheme of the algorithm, respectively. $y_t$ denotes the $t$-th job of $\pi$, and $p_{t,y_t}$ denotes the probability of the $t$-th job of $\pi$ is $y_t$.

Let $P_p$ represent the perturbation probability. In deviation from the conventional approach of selecting an optimal solution that minimizes the makespan, as proposed by (Corsini et al., 2024), we implement an alternative methodology for determining the suitable solution.

**Case 1:** If $\text{rand}(0, 1) \geq P_p$. Similar to (Corsini et al., 2024), we select the optimal solution as the pseudo-label.

**Case 2:** If $\text{rand}(0, 1) < P_p$. Instead of selecting the optimal solution, we randomly select one of the $\alpha$ parallel solutions (which are suboptimal) as the pseudo-label.

The original method anticipates that as the algotirhm learns over time, its predictions will progressively refine, with the pseudo-labels it generates increasingly approximating the true labels, ultimately converging upon them. However, this ideal is unattainable, as depicted in the schematic diagram in Figure 2 (a). For the sake of clarity, we shall assume that the algorithm requires only

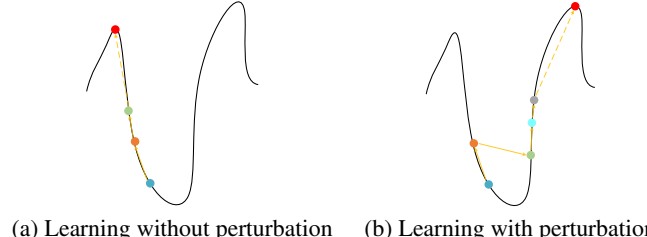

(a) Learning without perturbation    (b) Learning with perturbation

Figure 2: Illustration of the learning process. (a) illustrates the learning process without the influence of perturbations. initially at step 0, the algorithm is anchored at the blue point, using the orange point as the initial pseudo-label for learning. By step 1, the focus shifts to the green point, which the algorithm adopts as the new pseudo-label for further learning. This process continues, with the algorithm progressively converging towards a local optimum, represented by the red point. (b) illustrates the learning process with the influence of perturbations. With the introduction of perturbation at step 1, the algorithm is diverted from the peak associated with the local optimum. It opts for a green point, which is not the best option, but it is at the peak where the global optimum is located. As learning progresses, the algorithm moves towards the blue and grey points, ultimately settling at the globally optimal red point.

a single learning iteration to accomplish the acquisition of pseudo-labels. Initially, the algorithm gravitates toward a pseudo-label that approximates a local optimum. While this learning process enhances the predictive capability of algorithm for generating better pseudo-labels, it remains confined to the vicinity of the local optimum, failing to explore the broader landscape for a global optimum. Eventually, the pseudo-label tends to converge towards a local optimum, guiding the learning of algorithm in the same direction. This phenomenon is prevalent due to the solution space being a complex, multi-modal function, where it is nearly impossible to obtain solutions close to the global optimum within a single iteration (LeCun et al., 2015). Thus, we introduce a perturbation, as illustrated in Figure 2 (b). This perturbation enables the algorithm to escape the local optimum during the second learning phase, steering it towards peaks that are closer to the global optimum, and ultimately leading to convergence at the global optimum (true label). It is evident that the perturbation significantly enhances the probability of discovering the true label.

### 3.4 FURTHER ANALYSIS

Every COP can be envisioned as a decision-making process occurring on a graph (Korte et al., 2011), a foundational concept for the SS-FJSSP algorithm. This algorithm iteratively refines its decision-making skills on such graphs. The process is analogous to human decision-making: once the problem is understood, decisions are made incrementally, with each subsequent decision informed by the consequences of its predecessors to enhance accuracy. In this framework, comprehending the problem is the role of the encoder, considering the impact of previous decisions is the function of the state network, and the act of making decisions is the task of the decision network.

Next, the crux of the challenge is teaching the SS-FJSSP algorithm to learn the correct scheduling strategy, enabling it to identify the optimal solution once the problem is understood. This is a feat that humans are currently unable to achieve. The primary obstacle is the NP-Hard nature of the problem, which means an insufficient number of correct labels for the SS-FJSSP to learn from. To overcome this, we draw on the idea of genetic algorithm (Mitchell, 1998): instead of learning the true labels directly, we start with suboptimal labels and gradually approximate the true labels. The core of what makes this strategy work is that even though we do not know if a solution is a true label, for any two solutions, we can distinguish which of them is better (according to the magnitude of the fuzzy makespan).

Specifically, initially, the SS-FJSSP starts without any prior knowledge, randomly generating solutions as potential alternative labels. From these, the most optimal solution is chosen as the pseudo-label, allowing the algorithm to learn. Subsequently, armed with the knowledge acquired previously, the algorithm shows a tendency to predict better alternative labels, leading to the selection of an improved pseudo-label. Repeating this process, ideally, the true label will be selected as the pseudo-

label, and the algorithm will eventually learn to make decisions that correspond to it. However, the ideal scenario described above is so perfect that it is nearly unattainable in practice without introducing perturbations. The rationale behind this is that the aforementioned process is essentially akin to a genetic algorithm that employs only crossover without any mutation. Ideally, we anticipate that a superior solution would consistently generate new solutions that are at least as good as the existing ones, thus eventually leading to the global optimum. However, this ideal condition is met if and only if the solution space possesses particularly advantageous properties. In reality, the solution spaces of COPs or neural networks are often complex and multi-modal (LeCun et al., 2015). Consequently, we introduce a perturbation to emulate the role of mutation, which aids the algorithm in escaping local optima and identifying the global optimum, or the true label, as the pseudo-label.

Finally, it becomes evident that with minimal adjustments, this algorithm can be extended to other COPs. This adaptability stems from the fact that any COP can be effectively solved using GNNs once the problem is understood. In terms of decision-making, for any given COP, we are capable of determining the superior label, thereby approximating the true label as outlined in our strategy. Thus, this approach can be regarded as an innovative paradigm for addressing COPs, complementing existing methodologies such as mathematical methods (Ku & Beck, 2016), heuristic methods (Van Laarhoven et al., 1992), and reinforcement learning methods (Zhang et al., 2020).

## 4 NUMERICAL RESULTS AND COMPARISON

The SS-FJSSP algorithm is developed using Python 3.9 and PyTorch 1.3.1, and is executed on an Ubuntu 22.04 PC. The hardware setup includes an Intel Platinum 8358P processor and an NVIDIA GeForce RTX 4090 with 24GB of memory.

### 4.1 DATASET AND TEST INSTANCES

We randomly generated 30000 instances as the training set by following (Li et al., 2021). The size ($m \times n$) of the training set is $10 \times 10$, $15 \times 10$, $15 \times 15$, $20 \times 10$, $20 \times 15$, and $20 \times 20$, each with 5000 instances. The validation set is generated in the same way as the training set, except that the number of each size is 100. In order to test the performance of SS-FJSSP, we select 2 benchmarks: S (Sakawa & Mori, 1999; Sakawa & Kubota, 2000), and FT (Palacios et al., 2016). We also conducted the experiments for other popular benckmarks such as La (Palacios et al., 2016). The corresponding experimental results are deferred to Appendix.

### 4.2 ARCHITECTURE AND TRAINING

In the encoder, we utilize two GATs, each equipped with 3 attention heads and a LeakyReLU slope of 0.15. In $\text{GAT}_1$, the size of each head is set to 64 and their outputs are concatenated. In $\text{GAT}_2$, the size of each head is set to 128 and their outputs are averaged. Therefore, $h = 18 + 128 = 146$ and $e_i \in \mathbb{R}^{146}$. In the state network, the size of each head in MHA layer is set to 64 and their outputs are concatenated. This results in parameter matrices $\mathbf{W_1} \in \mathbb{R}^{11 \times 192}$ and $\mathbf{W_2} \in \mathbb{R}^{192 \times 128}$. Therefore, $d = 128$ and $s_i \in \mathbb{R}^{128}$. In the decision network, the FNN is implemented by a dense layer of 128 neurons, with a final layer of 1 neuron and Leaky-ReLU (slope = 0.15) activation function.

We utilize the Adam optimizer (Kingma, 2014) for training the SS-FJSSP algorithm. The training parameters are set as follows: the number of epochs to 30 and the learning rate to 0.0002. The batch size is configured at 16. Regarding the number of parallel solutions, denoted by $\alpha$, we set it to 128 during training and increase it to 2048 for testing. Additionally, the perturbation probability, $P_p$, is established at 0.05.

### 4.3 PERFORMANCE ON BENCKMARKS

In our evaluation, we have compared our SS-FJSSP with the state-of-the-art method in the field: Constraint Programming (CP), as detailed by (Afsar et al., 2023). As the code for CP is not open-source, our comparison is solely based on the results presented in (Afsar et al., 2023). Their experimental setup included the use of IBM ILOG CP Optimizer version 12.9 on a PC equipped with an Intel Xeon Gold 6132 processor, which operates at 2.6 GHz and is supported by 128 GB of RAM. The operating system utilized was Linux CentOS version 6.9.

Table 1: The comparison results of S-benchmark between SS-FJSSP and CP

| Size | Instance | SS-FJSSP | | CP | |
|---|---|---|---|---|---|
| | | FMS | RT | FMS | RT |
| 6 × 6 | S6.1 | **(56,80,103)** | **0.06** | **(56,80,103)** | 5 |
| | S6.2 | **(51,70,86)** | **0.06** | **(51,70,86)** | 2 |
| | S6.3 | **(51,65,84)** | **0.06** | **(51,65,84)** | 3 |
| | S6.4 | **(27,36,45)** | **0.07** | **(27,36,45)** | 3 |
| 10 × 10 | S10.1 | (93,135,137) | **0.21** | (96,129,161) | 36 |
| | S10.2 | (92,122,161) | **0.21** | (92,120,163) | 140 |
| | S10.3 | **(85,116,143)** | **0.21** | **(85,116,143)** | 43 |
| | S10.4 | **(28,47,64)** | **0.36** | **(28,47,64)** | 87 |

Table 2: The comparison results of FT-benchmark between SS-FJSSP and CP

| Size | Instance | SS-FJSSP | | CP | |
|---|---|---|---|---|---|
| | | FMS | RT | FMS | RT |
| 6 × 6 | FT06_F | **(54,55,56)** | **0.06** | **(54,55,56)** | 55.00 |
| | FT06_G | **(52,55,61)** | **0.06** | **(52,55,61)** | 55.75 |
| | FT06_L | **(44,57,73)** | **0.06** | **(44,57,73)** | 56.25 |
| | FT06_T | **(42,55,69)** | **0.06** | **(42,55,69)** | 55.25 |
| 10 × 10 | FT10_F | (917,967,1017) | **0.21** | **(882,930,989)** | 932.75 |
| | FT10_G | (892,962,1079) | **0.21** | **(865,930,1058)** | 945.75 |
| | FT10_S | (873,960,1074) | **0.20** | **(844,930,1047)** | 937.75 |
| 20 × 5 | FT20_F | (1157,1232,1307) | **0.33** | **(1094,1165,1238)** | 1165.50 |
| | FT20_G | (1129,1223,1141) | **0.33** | **(1074,1165,1356)** | 1190.00 |
| | FT20_T | (1176,1231,1288) | **0.33** | **(1112,1165,1216)** | 1164.50 |

Table 1 presents the experimental results on the S-benchmark, with FMS denoting the fuzzy makespan and RT signifying the running time. Notably, in six of the eight instances, our SS-FJSSP algorithm achieves a fuzzy makespan equivalent to that of CP, while significantly reducing the running time. Furthermore, in instances S10.1 and S10.2, the difference in fuzzy makespan between SS-FJSSP and CP is minimal, less than 3%, and the running time of SS-FJSSP is markedly superior, exceeding that of CP by more than 170 times. Table 2 illustrates the experimental results from the FT-benchmark, with notable findings highlighted. Specifically, in the 6 × 6 instances, the SS-FJSSP algorithm matches the fuzzy makespan of CP while operating at an impressive 900 times faster speed. For all other instances, the fuzzy makespan of SS-FJSSP exceeds that of CP by a maximum of 6%, and it runs at least 3000 times more quickly.

In conclusion, SS-FJSSP is a competitive algorithm despite the differences in experimental settings. Its significant advantage in terms of running time is mainly because it has effectively learned a collection of scheduling rules. For every new operation, SS-FJSSP merely needs to apply the rule once more, ensuring that the running time scales linearly with the complexity of problem. This is an advantage of learning-based approaches. On the other hand, methods such as CP solve problems through an iterative approach. As the problem size escalates, the solution space expands exponentially, a phenomenon known as combinatorial explosion. This exponential growth in the solution space typically results in prolonged running time for these methods.

## 5 CONCLUSION

In this paper, we introduce the SS-FJSSP, a self-supervised algorithm tailored to solve the FJSSP. Our in-depth analysis of its workings positions this approach as an innovative paradigm for addressing COPs. Exrensive experiments have demonstrated the effectiveness of the algorithm proposed in this paper. We believe that our proposed algorithm can be extended to other fuzzy scheduling problems, such as the fuzzy flow shop scheduling problem (Deng et al., 2023), and we regard this as a promising avenue for future research.

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

# APPENDIX FOR THE PERFORMANCE OF SS-FJSSP ON BENCHMARKS

## A PERFORMANCE ON LA-BENCHMARK

The La-benchmark, as introduced in (Palacios et al., 2016), is derived from the JSSP by applying various fuzzification techniques, which are denoted by the final letter of the sample identifier. The corresponding experimental outcomes are presented in Table 3, with a noteworthy upper limit of 21600 seconds set for the running time by the CP method.

The data clearly demonstrates that the SS-FJSSP algorithm possesses a significant advantage in running time, consistently surpassing the CP method by a factor of at least two orders of magnitude across all instances. This superiority is especially pronounced in complex scenarios, such as the LA21_F benchmark, where the SS-FJSSP's performance is significantly enhanced, reaching a remarkable four orders of magnitude faster. Moreover, it is worth highlighting that in almost one-third of the cases, SS-FJSSP matched CP in terms of fuzzy completion time. This not only highlights the algorithm's speed but also its effectiveness, showcasing its competitive edge in both aspects.

Table 3: The comparison results of La-benchmark between SS-FJSSP and CP

| Size | Instance | SS-FJSSP | | CP | |
| --- | --- | --- | --- | --- | --- |
| | | FMS | RT | FMS | RT |
| | La01_G | **(625,666,739)** | **0.10** | **(625,666,739)** | 9 |
| | La01_L | **(473,666,862)** | **0.10** | **(501,666,862)** | 9 |
| 10 × 5 | La02_S | (614,677,743) | **0.10** | (601,655,713) | 41 |
| | La03_G | (557,604,706) | **0.10** | (549,597,708) | 36 |
| | La05_G | **(548,593,665)** | **0.10** | **(548,593,665)** | 11 |
| | La06_L | **(667,926,1186)** | **0.34** | **(667,926,1186)** | 76 |
| 15 × 5 | La07_G | **(821,890,1003)** | **0.19** | **(821,890,1003)** | 18 |
| | La09_G | **(869,951,1065)** | **0.19** | **(869,951,1065)** | 20 |
| | La11_F | **(1164,1222,1280)** | **0.33** | **(1164,1222,1280)** | 41 |
| | La12_F | **(975,1039,1103)** | **0.33** | **(975,1039,1103)** | 88 |
| | La12_G | **(968,1039,1204)** | **0.33** | **(968,1039,1204)** | 104 |
| 20 × 5 | La13_F | **(1072,1150,1326)** | **0.33** | **(1072,1150,1228)** | 86 |
| | La13_G | **(1070,1150,1326)** | **0.33** | **(1070,1150,1326)** | 102 |
| | La14_F | **(1203,1292,1381)** | **0.33** | **(1203,1292,1381)** | 106 |
| | La14_G | **(1197,1292,1522)** | **0.33** | **(1197,1292,1522)** | 118 |
| 10 × 10 | La19_S | (765,856,941) | **0.21** | (752,842,948) | 1097 |
| | La20_Z | (821,916,1053) | **0.21** | (816,902,1036) | 394 |
| | La21_F | (1046,1122,1198) | **0.39** | (977,1046,1128) | 5187 |
| | La21_S | (997,1031,1264) | **0.39** | (943,1046,1167) | 4123 |
| | La21_Z | (1010,1123,1285) | **0.39** | (943,1046,1196) | 2427 |
| 15 × 10 | La22_Z | (856,965,1100) | **0.39** | (833,927,1065) | 282 |
| | La24_F | (911,983,1055) | **0.39** | (871,937,1010) | 2608 |
| | La24_S | (878,983,1089) | **0.39** | (840,938,1053) | 1533 |
| | La25_F | (954,1016,1078) | **0.39** | (913,978,1045) | 4223 |
| | La25_S | (926,1029,1154) | **0.39** | (882,977,1111) | 18561 |
| | La27_F | (1226,1321,1416) | **0.67** | (1154,1235,1316) | 6196 |
| 20 × 10 | La27_S | (1195,1330,1437) | **0.67** | (1132,1248,1366) | 21600 |
| | La29_F | (1146,1237,1328) | **0.67** | (1081,1154,1238) | 21600 |
| | La29_S | (1162,1311,1457) | **0.67** | (1052,1171,1319) | 21600 |
| | La36_S | (1176,1305,1443) | **0.59** | (1160,1275,1416) | 2931 |
| | La37_S | (1351,1499,1625) | **0.59** | (1262,1399,1560) | 979 |
| | La38_F | (1187,1272,1357) | **0.59** | (1128,1196,1284) | 21600 |
| 15 × 15 | La38_S | (1171,1276,1433) | **0.59** | (1091,1205,1342) | 21600 |
| | La39_S | (1162,1311,1457) | **0.59** | (1112,1233,1387) | 825 |
| | La40_F | (1199,1285,1371) | **0.59** | (1146,1224,1313) | 1013 |
| | La40_S | (1177,1290,1406) | **0.59** | (1119,1228,1357) | 3458 |

Table 4: The comparison results of the benchmarks of Lei and LP between SS-FJSSP and CP

| Size | Instance | SS-FJSSP | | CP | |
|---|---|---|---|---|---|
| | | FMS | RT | FMS | RT |
| 15 × 10 | Lei01 | (145,211,280) | **0.39** | **(136,200,259)** | 9977 |
| | Lei02 | (129,178,226) | **0.55** | **(118,164,214)** | 234 |
| 16 × 16 | LP01 | (150,196,250) | **0.69** | **(145,190,246)** | 21600 |

Table 5: The comparison results of the benchmarks of ABZ and ORB between SS-FJSSP and CP

| Size | Instance | SS-FJSSP | | CP | |
|---|---|---|---|---|---|
| | | FMS | RT | FMS | RT |
| 10 × 10 | ABZ5_Z | (1139,1253,1434) | **0.21** | **(1111,1239,1414)** | 876 |
| | ABZ6_G | (882,952,1118) | **0.20** | **(876,943,1068)** | 287 |
| | ABZ6_Z | (858,954,1097) | **0.21** | **(842,945,1085)** | 363 |
| | ORB01_Z | (1036,1146,1327) | **0.34** | **(961,1060,1215)** | 1415 |
| | ORB02_Z | (803,901,1023) | **0.45** | **(784,889,1022)** | 507 |
| | ORB03_Z | (962,1082,1251) | **0.30** | **(900,1005,1151)** | 1218 |
| | ORB04_Z | (961,1080,1233) | **0.30** | **(894,1006,1154)** | 238 |
| | ORB05_Z | (823,914,1049) | **0.31** | **(800,887,1018)** | 260 |
| 20 × 15 | ABZ7_F | (660,710,760) | **1.00** | **(628,660,703)** | 21600 |
| | ABZ8_F | (682,723,764) | **1.00** | **(645,681,726)** | 21600 |
| | ABZ9_F | (719,758,797) | **1.15** | **(668,704,758)** | 21600 |

## B    PERFORMANCE ON THE BENCHMARKS OF LEI AND LP

The Lei-benchmark (Lei, 2010b) and the LP-benchmark (Li & Pan, 2013) are specifically developed for fuzzy scheduling purposes, rather than being derived from fuzzifying crisp problems. These benchmarks, despite their compact size, offer a high degree of complexity, often requiring considerable computational resources for the CP method to find solutions. Although the SS-FJSSP algorithm does not always match the solution quality of CP, it provides nearly identical solutions with remarkable efficiency. The experimental findings are detailed in Table 4.

The slightly inferior solution quality produced by SS-FJSSP may be due to the fact that the training set, which is randomly generated, does not fully capture the intricacies of these carefully designed instances. It is expected that with an improved training set that more accurately reflects the complexity of these benchmarks, SS-FJSSP will be able to produce more satisfactory results.

## C    PERFORMANCE ON THE BENCHMARKS OF ABZ AND ORB

Both the ABZ-benchmark (Palacios et al., 2016) and the ORB-benchmark (Zheng et al., 2011) originate from the fuzzification of highly intricate original problems, retaining their inherent complexity in the fuzzified versions. As depicted in Table 5, while no instance of SS-FJSSP have been able to match the results of CP, the discrepancy is minimal, and the SS-FJSSP still holds a significant advantage in terms of running time. The suboptimal performance in solution quality is also likely attributable to the training set's insufficient complexity, which fails to encapsulate the full intricacy of these benchmarks.

## D    PERFORMANCE ON THE TA-BENCHMARK

The Ta-benchmark (Afsar et al., 2023), derived from a fuzzy approach, comprises an extensive collection of instances, totaling 80. The corresponding experimental results are meticulously detailed across two tables: Table 6 and Table 7. These results corroborate the characteristics previously discussed, offering further insight into the performance of the evaluated methods.

Table 6: The comparison results of Ta-benchmark between SS-FJSSP and CP in Part I

| Size | Instance | SS-FJSSP | | CP | |
|---|---|---|---|---|---|
| | | FMS | RT | FMS | RT |
| 15 × 15 | Ta01_F | (1204,1336,1468) | **0.89** | **(1149,1231,1344)** | 450 |
| | Ta02_F | (1220,1313,1406) | **0.90** | **(1152,1245,1342)** | 1562 |
| | Ta03_F | (1196,1295,1394) | **0.89** | **(1147,1219,1307)** | 2099 |
| | Ta04_F | (1145,1226,1307) | **0.89** | **(1083,1175,1275)** | 1831 |
| | Ta05_F | (1201,1290,1379) | **0.89** | **(1158,1224,1319)** | 21600 |
| | Ta06_F | (1195,1281,1367) | **0.89** | **(1174,1246,1335)** | 21600 |
| | Ta07_F | (1181,1284,1387) | **0.89** | **(1147,1228,1322)** | 15011 |
| | Ta08_F | (1201,1285,1396) | **0.89** | **(1134,1218,1327)** | 18619 |
| | Ta09_F | (1245,1370,1495) | **0.89** | **(1181,1274,1393)** | 11445 |
| | Ta10_F | (1206,1317,1428) | **0.89** | **(1163,1243,1326)** | 1882 |
| 20 × 15 | Ta11_F | (1337,1460,1583) | **1.62** | **(1290,1387,1488)** | 21600 |
| | Ta12_F | (1349,1464,1579) | **1.62** | **(1279,1377,1488)** | 21600 |
| | Ta13_F | (1360,1452,1544) | **1.62** | **(1312,1413,1536)** | 21600 |
| | Ta14_F | (1323,1409,1495) | **1.62** | **(1248,1345,1442)** | 1544 |
| | Ta15_F | (1328,1452,1576) | **1.62** | **(1279,1373,1470)** | 21600 |
| | Ta16_F | (1355,1480,1605) | **1.62** | **(1283,1381,1492)** | 21600 |
| | Ta17_F | (1474,1582,1690) | **1.61** | **(1352,1464,1593)** | 1468 |
| | Ta18_F | (1411,1538,1665) | **1.62** | **(1367,1474,1593)** | 21600 |
| | Ta19_F | (1351,1446,1541) | **1.62** | **(1250,1378,1507)** | 21600 |
| | Ta20_F | (1346,1456,1566) | **1.62** | **(1278,1368,1471)** | 21600 |
| 20 × 20 | Ta21_F | (1672,1792,1912) | **2.16** | **(1596,1697,1824)** | 21600 |
| | Ta22_F | (1589,1731,1873) | **2.16** | **(1546,1666,1809)** | 21600 |
| | Ta23_F | (1589,1708,1827) | **2.17** | **(1526,1614,1726)** | 21600 |
| | Ta24_F | (1663,1788,1913) | **2.16** | **(1579,1708,1847)** | 21600 |
| | Ta25_F | (1602,1718,1834) | **2.16** | **(1524,1632,1765)** | 21600 |
| | Ta26_F | (1640,1769,1898) | **2.30** | **(1643,1742,1865)** | 21600 |
| | Ta27_F | (1736,1844,1952) | **2.16** | **(1664,1780,1914)** | 21600 |
| | Ta28_F | (1611,1733,1855) | **2.17** | **(1561,1662,1770)** | 21600 |
| | Ta29_F | (1593,1726,1859) | **2.16** | **(1554,1660,1780)** | 21600 |
| | Ta30_F | (1593,1726,1859) | **2.17** | **(1545,1647,1756)** | 21600 |
| 30 × 15 | Ta31_F | (1794,1944,2094) | **3.26** | **(1682,1805,1938)** | 21600 |
| | Ta32_F | (1872,1994,2116) | **3.27** | **(1690,1824,1979)** | 21600 |
| | Ta33_F | (1859,2006,2153) | **3.27** | **(1706,1853,2008)** | 21600 |
| | Ta34_F | (1858,1995,2132) | **3.26** | **(1709,1866,2023)** | 21600 |
| | Ta35_F | (1895,2067,2239) | **3.27** | **(1847,2007,2167)** | 2367 |
| | Ta36_F | (1859,2000,2141) | **3.26** | **(1720,1847,1987)** | 21600 |
| | Ta37_F | (1837,1972,2107) | **3.26** | **(1647,1796,1967)** | 21600 |
| | Ta38_F | (1692,1839,1986) | **3.27** | **(1575,1692,1847)** | 21600 |
| | Ta39_F | (1810,1963,2116) | **3.27** | **(1711,1822,1933)** | 21600 |
| | Ta40_F | (1743,1876,2009) | **3.27** | **(1600,1714,1858)** | 21600 |

Table 7: The comparison results of Ta-benchmark between SS-FJSSP and CP in Part II

| Size | Instance | SS-FJSSP | | CP | |
|---|---|---|---|---|---|
| | | FMS | RT | FMS | RT |
| 30 × 20 | Ta41_F | (2169,2311,2453) | 0.89 | (1958,2090,2222) | 21600 |
| | Ta42_F | (2011,2167,2323) | 0.90 | (1862,2032,2205) | 21600 |
| | Ta43_F | (1936,2085,2234) | 0.89 | (1809,1949,2090) | 21600 |
| | Ta44_F | (2103,2219,2335) | 0.89 | (1933,2044,2173) | 21600 |
| | Ta45_F | (2058,2191,2324) | 0.89 | (1928,2036,2170) | 21600 |
| | Ta46_F | (2087,2230,2373) | 0.89 | (1949,2066,2215) | 21600 |
| | Ta47_F | (1959,2087,2215) | 0.89 | (1828,1973,2120) | 21600 |
| | Ta48_F | (2036,2199,2362) | 0.89 | (1881,2012,2160) | 21600 |
| | Ta49_F | (2040,2175,2310) | 0.89 | (1897,2030,2177) | 21600 |
| | Ta50_F | (2061,2221,2381) | 0.89 | (1902,2032,2172) | 21600 |
| 50 × 15 | Ta51_F | (2804,3052,3300) | 1.62 | (2549,2760,2971) | 21600 |
| | Ta52_F | (2696,2942,3188) | 1.62 | (2581,2756,2931) | 21600 |
| | Ta53_F | (2628,2852,3076) | 1.62 | (2521,2717,2915) | 21600 |
| | Ta54_F | (2628,2863,3098) | 1.62 | (2605,2839,3073) | 21600 |
| | Ta55_F | (2726,2935,3144) | 1.62 | (2497,2679,2880) | 21600 |
| | Ta56_F | (2707,2921,3135) | 1.62 | (2574,2781,2994) | 21600 |
| | Ta57_F | (2852,3097,3342) | 1.61 | (2719,2943,3167) | 21600 |
| | Ta58_F | (2777,3025,3273) | 1.62 | (2682,2885,3128) | 21600 |
| | Ta59_F | (2683,2900,3117) | 1.62 | (2469,2655,2865) | 21600 |
| | Ta60_F | (2614,2828,3042) | 1.62 | (2525,2723,2927) | 21600 |
| 50 × 20 | Ta61_F | (2857,3099,3341) | 2.16 | (2691,2868,3072) | 21600 |
| | Ta62_F | (2921,3144,3367) | 2.16 | (2729,2904,3085) | 21600 |
| | Ta63_F | (2745,2973,3201) | 2.17 | (2529,2755,2987) | 21600 |
| | Ta64_F | (2601,2823,3045) | 2.16 | (2495,2702,2909) | 21600 |
| | Ta65_F | (2795,3013,3231) | 2.16 | (2559,2725,2948) | 21600 |
| | Ta66_F | (2813,3040,3267) | 2.30 | (2625,2845,3094) | 21600 |
| | Ta67_F | (2753,2994,3235) | 2.16 | (2606,2826,3083) | 21600 |
| | Ta68_F | (2657,2886,3115) | 2.17 | (2609,2784,2971) | 21600 |
| | Ta69_F | (2992,3233,3474) | 2.16 | (2850,3071,3292) | 21600 |
| | Ta70_F | (2991,3273,3555) | 2.17 | (2792,2995,3219) | 21600 |
| 100 × 20 | Ta71_F | (5263,5613,5963) | 3.26 | (5089,5464,5839) | 21600 |
| | Ta72_F | (4901,5264,5627) | 3.27 | (4822,5181,5540) | 21600 |
| | Ta73_F | (5392,5777,6162) | 3.27 | (5195,5568,5941) | 21600 |
| | Ta74_F | (4972,5351,5730) | 3.26 | (4950,5339,5739) | 21600 |
| | Ta75_F | (5342,5710,6078) | 3.27 | (4959,5392,5830) | 21600 |
| | Ta76_F | (5175,5560,5945) | 3.26 | (5022,5342,5736) | 21600 |
| | Ta77_F | (5128,5462,5796) | 3.26 | (5050,5436,5822) | 21600 |
| | Ta78_F | (5137,5498,5859) | 3.27 | (4980,5394,5808) | 21600 |
| | Ta79_F | (5006,5392,5778) | 3.27 | (4974,5358,5744) | 21600 |
| | Ta80_F | (4922,5341,5760) | 3.27 | (4833,5183,5533) | 21600 |

