# OpenReview forum: "Solving the Fuzzy Job Shop Scheduling Problem via Learning Approaches"
_ICLR.cc/2025/Conference — Submitted to ICLR 2025_

### Official Review · Reviewer_yP6m · 2024-10-25

**Soundness:** 2
**Presentation:** 2
**Contribution:** 1
**Rating:** 3
**Confidence:** 3

**Summary:**

This paper addresses the Fuzzy Job Shop Scheduling Problem (FJSSP), an extension of the traditional job shop scheduling problem that incorporates uncertainty, better reflecting real-world manufacturing environments. The authors proposed a self-supervised algorithm (SS-FJSSP) that employs neural combinatorial optimization to handle fuzzy data and solve FJSSP. The method is computationally efficient, achieving comparable results to state-of-the-art algorithms on several benchmark problems.

**Strengths:**

The integration of neural networks with fuzzy scheduling and self-supervised learning seems to be interesting and important.

The paper is generally well-structured, and the explanations of both the problem and the proposed solution are clear.

The proposed solution addresses a critical issue in fuzzy scheduling by improving computational efficiency while maintaining accuracy.

**Weaknesses:**

My primary concern about this paper is regarding its novelty. While using neural combinatorial optimization (NCO) techniques to solve FJSSP problems might be novel, the high-level graph representation of the FJSSP problem adopted in this paper is very similar to many existing studies. It seems that the key difference is the introduction of fuzzy model related features and feature vectors. However, the technical novelty associated with these features has not been properly highlighted and justified.

Meanwhile, the training strategy does not seem to be new. Perhaps random solution selection is a novel element of the training algorithm. Nevertheless, the effectiveness of this selection mechanism is only discussed intuitively. More thorough theoretical and empirical analysis may be necessary to clearly quantify its importance for the newly proposed NCO system to find near-optimal solutions. To my understanding, the analogy with genetic algorithms does not reveal the true advantages of using random solution selection techniques, since the proposed NCO system is not closely related to genetic algorithms.

The baseline algorithms adopted in this paper were published about 8 to 9 years ago. It is important to include more recent baselines in the experimental comparison to truly understand the technical advancement introduced by the new approach. In particular, to clearly demonstrate the advantages of the newly developed encoder-decoder network architecture, several existing approaches using different GNN architecture designs should be experimentally examined and compared in this paper.

**Questions:**

Could you provide more detail on the technical novelty of the fuzzy model-related features and the feature vectors introduced in your method? How do these features fundamentally differ from existing studies that use similar high-level graph representations for job shop scheduling or similar problems?

The random solution selection mechanism is mentioned as part of your training strategy. Can you offer a more thorough explanation, both theoretically and empirically, to justify the effectiveness of this technique? Specifically, how does this random solution selection improve performance compared to existing methods, and how does it contribute to the near-optimal solutions?

The baseline algorithms used for comparison are quite dated (8-9 years old). Are there any recent state-of-the-art methods, particularly involving newer GNN architectures, that could be included in the experiments? How would incorporating these newer baselines demonstrate the technical advancements of your approach?

Have you considered other recent GNN-based architectures for job shop scheduling or other related problems? How does your approach specifically improve on those?

**Details Of Ethics Concerns:**

I do not have ethics concerns.

---

### Official Review · Reviewer_hhYx · 2024-10-29

**Soundness:** 1
**Presentation:** 2
**Contribution:** 1
**Rating:** 3
**Confidence:** 5

**Summary:**

This paper proposed a self-supervised algorithm for FJSSP (SS-FJSSP) that uses an iterative mechanism to refine pseudo-labels, moving from suboptimal to optimal solutions. Meanwhile, this algorithm effectively bypasses the common neural combinatorial optimization challenge of obtaining true labels. This paper addresses this gap by examining the potential of neural networks to process fuzzy information for solving FJSSP.

**Strengths:**

The paper proposed a deep network to address addresses this gap by examining the potential of neural networks to process fuzzy information for solving FJSSP.

**Weaknesses:**

The authors just compared their method with one method, which is not enough to verify the effectiveness of the proposed method. There are state-of-the-art evolutionary computation algorithms for FJSSP. What is the advantage of the proposed method over them? The comparison between the proposed method and them should be implemented to analyze the difference.  Meanwhile, the proposed algorithm is not very competitive in items of FMS, and its performance is poorer than that of the comparison algorithm in some problems.
The paper lacks parameter analysis and ablation study, making a gap between the results and the conclusion.

**Questions:**

1. This paper lacks contributions and originality. It just proposed a network based on an encoder and decoder for FJSSP. The encoder-decoder architecture is common in a deep learning network. Therefore, the author should discuss how to design the encoder and the decoder in detail.
2. The parameter analysis and architecture of the proposed should be further discussed. The paper uses GAT as the backbone. Is it possible to use another network to solve the problem?
3. The effectiveness of the proposed method is questionable. As shown in Table 2, the results obtained from SS-FJSSP are not significantly different from those obtained with CP.
4. The paper lacks parameter analysis and ablation study, making a gap between the results and the conclusion.
5. The comparison of RT is not fair. SS-FJSSP and CP are not executed in the same environment. The training time of SS-FJSSP should be included in the final RT. Additionally, SS-FJSSP uses GPUs to accelerate computation, while CP is just implemented on a CPU.

---

### Official Review · Reviewer_jaaE · 2024-10-30

**Soundness:** 2
**Presentation:** 2
**Contribution:** 2
**Rating:** 3
**Confidence:** 4

**Summary:**

A semi-supervised learning approach for the fuzzy job shop scheduling problem (FJSSP) is presented. Experimental results are provided on a subset of the standard instances for this problem and compared against a constraint programming baseline.

**Strengths:**

The paper organized well, but contains some really strange text at times. I find it hard to believe the line "The surge in deep learning has catalyzed the emergence of neural combinatorial optimization (NCO) sparking a burgeoning interest in leveraging..." was not generated by an LLM (note that I do not believe the paper was written by an LLM, just this part). Regardless of where it comes from, it is not so good. The document is overall understandable, the typos do not impact this. It would be best, though, if the authors used GPT to support fixing typos, or just watch the spellchecker (e.g., 'algotirhm' on page 6)

The main novelty is the application of the semi-supervised learning approach from Corsini et al. (2024) (note that this paper is cited with its arxiv version, but actually it is accepted to NeurIPS 24) to the FJSSP with a few twists. There are some interesting ideas in here. The semi-supervised approach is particularly interesting, although the novelty in this paper is (1) only a small advancement over the previous work of Corsini et al. and (2) not examined in an ablation analysis at all (see below). The second claim to novelty about a "refinement process" seems to me to just be the Corsini paper again, so I am not sure what is meant here -- the perturbation technique?

**Weaknesses:**

There are two major issues with the paper that right now prevent me from being more positive about it.

1. The new perturbation strategy is not experimentally validated at all. First note that it is said that "Corsini et al. (2024) ... select the optimal solution as the pseudo-label". My understanding of the Corsini paper is that they use the best solution from the sample. I assume that this is just a typo though. More importantly, where does P_p come from and who sets it? Why is this value not experimentally validated? The experimental section says that it is set to 0.05 and that's all. First, turning it off is an important ablation step, and showing the sensitivity is also very interesting. I also wonder why the authors do not try this technique out on the original JSSP? It would improve the generalizability of this paper, which is rather sharply focused on the FJSSP right now and, in all honesty, ICLR is not an applications conference. The reviewers need to at least believe there is a chance of generalization here, but right now there is no evidence.
2. The experimental results are not complete. They compare the approach with a CP model, mostly lose, and then claim victory because it is faster. First, CP is proving optimality and this method is not, so it is a completely apples to oranges comparison. Second, where do these results even come from in the Afsar et al. (2023) paper? I do not see these runtimes; on the contrary, Table 5 in the Afsar paper indicate that S6.1-4 only need 0.1 seconds to solve to optimality and the heuristic approach HTS only 0.2 seconds. On S10.1-4 on average 76 seconds are needed for the CP model and 1.0 for HTS. This brings me to a question: where is HTS in this work? How does HTS compare to the SS-FJSSP approach? I note that this comparison is quite important as SS-FJSSP has many domain specific components in it, so the method is not interesting in and of itself -- it ought to be as good as the current techniques available. One more note here: the authors make note that they do not re-implement the CP model because the source code is not available... but we are talking about a model with 5 constraints, this would not be much of a challenge...

I note that I am also a bit wary about the FJSSP in general. The fuzzy literature makes rather simplifying assumptions (triangular distribution), thus avoiding any of the hard parts of modeling stochastic optimization problems while still claiming to be more realistic. While I agree that it is more realistic than a standard JSSP, the question is whether it is really worth the trouble compared to just modeling a two-stage, robust or chance constrained problem. My opinion is that it is not because the triangular distribution is unrealistic for a variety of reasons (real distributions are often shifted, can have long tails, etc.). Thus, I think the authors are betting all their money here on a rather weak application that is not at the level of top conferences.


Some other notes about the paper for the authors to improve:
1. The references in the introduction about the FJSSP are not terribly convincing that it is an important problem. For a paper at a top conference, I would expect papers at top conferences (or an argument why those venues have overlooked this important problem)
2. The second paragraph of the introduction is not well written; it is very long and just meanders through the literature.
3. Page 3: Additional should be addition
4. The explanation about the arrow operator in 2.2 is unclear (namely i->x or x->j) -- I think this can be interpreted (incorrectly) as banning chains of operations.
5. The mathematical model ought to have a min before the max so it is more clear. Also the math model does not use the max operator in the constraints, which seems like it ought to be necessary? I admit I am not so familiar with modeling with fuzzy variables, so maybe there is a convention there I am not aware of.
6. The jobs numbers ought to be more clear in Figure 1 so it is easier to understand the scheduling scheme.

**Questions:**

1. Why isn't P_p the focus of a sensitivity analysis?
2. Why isn't HTS compared against? Where do the runtimes for CP come from? Why should I care that this method is faster than an exact method?

---

### Official Review · Reviewer_2yZo · 2024-11-03

**Soundness:** 3
**Presentation:** 3
**Contribution:** 2
**Rating:** 5
**Confidence:** 3

**Summary:**

This paper proposes a self-supervised algorithm for the FJSSP (SS-FJSSP) that employs an iterative mechanism to refine pseudo-labels, progressively transitioning from suboptimal to optimal solutions. The proposed method showed better performance than the constrained programming methods on the FJSSP problems.

**Strengths:**

1.A new problem of JSSP is investigated and solved, considering the fuzzy processing time.

2. The proposed method performs better than the compared method.

3. The proposed method can achieve self-learning by using pseudo labels for training.

4. The idea is good and the paper is well-written.

**Weaknesses:**

1. The experiments are not sufficient to show the contributions of the proposed method without comparing other methods.

2. The novelty is limited since the fuzzy job shop scheduling problem has been investigated and the paper looks like an extension of existing work by adding the fuzzy processing time.

**Questions:**

1. The proposed method is not compared with state-of-the-art methods, including heuristic methods and reinforcement learning methods.

2. Some arguments are not true or correct, particularly the argument related to FJSSP and the dynamic or fuzzy processing time. The JSSP problems have been investigated for years and the dynamic ones, or the fuzzy ones have also been solved by using many other algorithms, while the paper totally ignored existing work.

3. The parameter setting is not justified. Or ablation analysis on parameter settings is needed.

4. Please clarify that the running time of the proposed algorithm does not include the training time of the neural networks.

5. I am wondering whether the compared method, CP also use a large training set to train a model and then is tested on the test set. If it is not, the comparison will not be fair.

6. The analysis of learning with perturbation is desirable. How it is achieved to achieve the goal shown in Fig. 2. It is also necessary to discuss its pros. and cons.

7. Since JSSP problems do not have true labels, the reinforcement learning method is another learning method commonly used for this problem. How is the proposed method compared with the reinforcement learning method?

---

### Meta-Review · Area_Chair_pZnZ · 2024-12-20

**Metareview:**

This paper proposed a learning approach to solve the fuzzy job shop scheduling problem. It is based on an existing self-supervised learning mechanism, with an iterative refinement process to incrementally refine the labels. Reviewers acknowledged the written and organization of this paper. However, they raised serious concerns including 1) limited novelty since the neural architecture and self-supervised training mechanism resembles much to existing works; 2) lack of motivation of studying the fuzzy JSP as well as its practical value; and 3) incomplete evaluation with missing SOTA baselines and ablation study. Considering that all reviewers are negative, I recommend rejection.

**Additional Comments On Reviewer Discussion:**

Authors did not provide responses to reviewers' comments.

---

### Decision · Program_Chairs · 2025-01-22

Reject